

# Vaccine refusal in pregnant women in Kahramanmaraş: a community-based study from Türkiye

Ramazan Azim Okyay[1], Erhan Kaya[2] and Ayşegül Erdoğan[2]

[1] Sarıçam District Health Department, Adana, Türkiye
[2] Faculty of Medicine, Department of Public Health, Kahramanmaraş Sütçü İmam University, Kahramanmaraş, Türkiye

## ABSTRACT

**Background:** The global health landscape is increasingly challenged by the rejection of childhood vaccines. This study investigates vaccination reservations and refusal among pregnant women.

**Methods:** A cross-sectional study was conducted in Kahramanmaraş, Türkiye, spanning January to March 2019, entailing face to face interviews with 938 pregnant women. The questionnaire collected sociodemographic data and probed vaccination attitudes. The decision to vaccinate the baby was treated as the dependent variable, analyzed alongside sociodemographic factors and other variables. Data underwent evaluation *via* descriptive analysis, the Chi-square test, and binary logistic regression.

**Results:** Participants had a mean age of 27.6 years. Among them 20 (2.1%) expressed intent to either abstain from or partially vaccinate their babies, with 70% citing apprehensions regarding potential vaccine-related harm. Statistical analysis unveiled that higher economic income, elevated parental education level, fewer than two children, reliance on the Internet for vaccination information, and a lack of trust in physicians significantly correlated with vaccine refusal among pregnant women ($p < 0.05$).

**Conclusions:** The study concludes that dissemination of vaccination information by the healthcare professionals, complemented by the enactment of pro-vaccine internet policies holds promise in shaping vaccination behavior.

Corresponding author
Ramazan Azim Okyay,
razim01@gmail.com

## INTRODUCTION

The Centers for Disease Control and Prevention (CDC) ranks vaccination and infectious disease prevention among the last century's ten greatest public health achievements (*Centers for Disease Control and Prevention (CDC), 2011*). To ensure and maintain community immunity, high vaccination rates are required. Vaccine hesitancy is emerging as a growing concern, as it is a major reason for community immunization to shrink rapidly in recent years worldwide. Vaccine hesitancy is defined as a refusal or delay in accepting vaccines despite the availability of vaccination services, and it occurs in a changing and complex context. The Strategic Advisory Group of Experts (SAGE) Working

Group emphasizes the importance of categorizing the factors that influence vaccine hesitancy and acceptance (*The SAGE Vaccine Hesitancy Working Group, 2014*, *2017*; *MacDonald, 2015*). The World Health Organization (WHO) has identified vaccine hesitancy as one of the ten global health problems for 2019, noting that the number of countries reporting data on vaccine hesitancy has been increasing since 2014 (*World Health Organization (WHO), 2019*, *2023*). In the 2018 WHO report, it was stated that global immunization prevented 2–3 million deaths per year in the world and that approximately 19.4 million children under the age of one were not vaccinated. In addition, the WHO estimates that a total of 1.5 million deaths in all age groups could be prevented if vaccine coverage increased (*The SAGE Vaccine Hesitancy Working Group, 2018*).

Within the scope of Türkiye's Extended Immunization Program, primary healthcare personnel provide routine vaccination services to children free of charge. According to this program, children are vaccinated against pertussis, diphtheria, tetanus, measles, rubella, mumps, tuberculosis, poliomyelitis, influenza type b, pneumococcus, hepatitis a, b, and varicella. Disease control, elimination and eradication programs are carried out to reduce the morbidity and mortality of these diseases. In Türkiye, the targets of polio eradication were achieved in 2002, and the targets of maternal and neonatal tetanus elimination were achieved in 2009 (*Gür, 2019*). However, despite these successes, if unvaccinated or under-vaccinated children accumulate due to vaccine refusal or vaccine hesitation, some vaccine-preventable infectious diseases may increase in the future (*Omer et al., 2009*). Some studies reported that the idea of vaccination rejection or acceptance of the vaccine starts during the pregnancy period, and the importance of interventions for vaccination hesitation during this period is emphasized (*Glanz et al., 2013*; *Danchin et al., 2018*).

Limited studies on this subject were observed in our literature review. Therefore, this study aimed to investigate the frequency and factors affecting vaccine refusal in pregnant women who will soon have a baby.

## MATERIALS AND METHODS

### Study design

This study was conducted in the Kahramanmaraş province in Türkiye's East Mediterranean region. The city center consists of two districts called Dulkadiroğlu and Onikişubat. The study was planned as a cross-sectional design. Since it was impractical to reach all pregnant women in the city, we calculated a sample size that would be representative of the general population.

### Sampling and data collection

The number of live births in Dulkadiroğlu and Onikişubat Districts of Kahramanmaraş in 2017 were 4,469 and 7,973, respectively. It was predicted that there would be approximately 13,000 pregnancies in Kahramanmaraş in 2018. With this in hand, to improve generalizability, the aim was to reach the maximum sample size, by assuming the prevalence of vaccine refusal to be 50%. Finally, the number of pregnant women to be included in the study was calculated as 987 in 95% confidence interval and with a margin of error of 3% in Kahramanmaraş region. The calculated sample size was distributed

by weighting according to the districts' populations. As part of primary health care in Türkiye, registered pregnant women are monitored in family health centers. For this reason, the sample was planned to be collected in randomly selected family health centers. At the time of the study there were 24 family health centers in Dulkadiroğlu and 37 family health centers in Onikişubat. Four family health centers from Dulkadiroğlu and seven from Onikişubat were randomly selected.

In family health centers, a questionnaire form based on the literature was applied to pregnant women through face-to-face interviews. When creating the questionnaire, existing research in the field of vaccine hesitancy or refusal was referred to for guidance. The misconceptions and concerns about vaccine safety outlined in previous studies helped shape the questions we included (*Daley et al., 2018*; *Danchin et al., 2018*; *Dáňová et al., 2015*; *Giambi et al., 2018*; *Massimi et al., 2017*). The questionnaire consists of 24 questions. The first 15 questions inquire about the participants' sociodemographic characteristics (such as age, nationality, pregnancy status, number of children, education level, income, *etc.*). The remaining nine questions evaluate the participants' knowledge, attitudes, and behaviors regarding childhood vaccinations. These questions cover topics such as the vaccination status of their children, perceived benefits and risks of vaccinations, and the necessity and reliability of vaccinations, among others. In this way, the questionnaire encompasses the participants' demographic characteristics, views, and behaviors regarding childhood vaccinations.

The survey was conducted by resident physicians of Kahramanmaraş Sütçü İmam University Department of Public Health, between January 1 and March 31, 2019. Out of 970 (98.7% of calculated sample size) pregnant women reached, only 32 individuals (3.3%) declined to participate in the study. By the end of the survey, 938 pregnant women, (95.5% of calculated sample size), participated in the study.

## Statistical analysis

The data were analyzed in SPSS v15 package program. The conformity of the variables to the normal distribution was evaluated with the Shapiro-Wilk test. Descriptive statistics were expressed as mean, standard deviation, number and percentage. The Pearson chi-square test was used to determine the differences between categorical variables. Participants who stated that they would not or only partially vaccinate their unborn babies were considered as the vaccine refusal group; the remaining were called the pro-vaccine group. A binary regression model was established to asses which of the participants' thoughts regarding vaccination acted as independent factors regarding vaccine refusal. Thoughts showing statistically significant differences ($p < 0.001$) between the vaccine refusal group and the pro-vaccine group were included in the model, in addition to age and education level. Among thoughts that were highly correlated with each other, only one was included in the model to determine the most suitable model. The Hosmer-Lemeshow test was used to assess the model fit. Since the $p$ value of the logistic regression model's Hosmer-Lemeshow test was >0.05, it was considered a good model with high predictive value (−2 Log likelihood = 46.925, Cox & Snell R square = 0.096, Nagelkerke R

square = 0.670). All statistical analyses in our study were two-tailed. *P* < 0.05 was considered significant in statistical analysis.

## Ethical considerations

The study was performed according to the tenets of the Declaration of Helsinki for research involving human subjects. Participants were informed about the use and anonymization of data, ensuring the anonymity of each participant's survey responses. Participants could proceed to the next question of the questionnaire even if they failed to provide a response to a particular item. It was not mandatory to answer all questions to move forward in the survey. Participation in the study was purely voluntary and that participants did not receive any gifts or monetary compensation. In addition participants were informed that they could withdraw from the research at any time, with or without justification. The study was approved by the Ethics Committee of Kahramanmaraş Sütçü İmam University with the date of October 24, 2018 and number 21, and necessary permissions were obtained from the health authority to conduct the study at family health centers. Written consent was obtained from the participants before starting the study.

## RESULTS

A total of 938 pregnant women were included in the study. The mean age of the pregnant women participating in the study was 27.6 ± 5.3 (minimum = 17, maximum = 43). The educational status of the participants was as follows: 0.9% were illiterate, 1.4% were literate but did not attend school, 17.4% were primary school graduates, 24.8% were secondary school graduates, 32% were high school graduates, and 23.6% were university graduates or above. Additionally, 17.3% of the participants were working, and 68.2% had at least one child (Table 1).

In our study, 98.3% of the pregnant women stated that they had fully immunized their living child, and 97.9% stated they planned to immunize their future baby fully. Of the 20 pregnant women who stated that they would not vaccinate or partially vaccinate their baby, 14 (70%) stated that they would not vaccinate on the grounds that vaccines are harmful, and 14 (70%) of these pregnant women with vaccine refusal thoughts were women expecting their first child. Pregnant women reported receiving the most information about vaccination from health personnel (74.7%) and the Internet (12.2%). Additionally, 2.8% of pregnant women expressed distrust towards their physicians, and 1.7% were unsure about the location of vaccine administration (Table 2).

In statistical analyses, high economic income, high parental education level, having no living children, having fewer than two children, using the Internet as a source of information about vaccination and not trusting the physician significantly increased the thought of vaccine hesitation and refusal in pregnant women (*p* < 0.05) (Table 3).

In our study, pregnant women were asked about their knowledge and thoughts about vaccination. The answers given by the pregnant women to these questions and the relationship analysis between these answers and their hesitation-refusal thoughts are shown in Table 4.

**Table 1 Participitans' sociodemographic information and pregnancy-related characteristics.**

| | | n | (%) |
|---|---|---|---|
| Age group (years) (n = 933) | 29 and below | 610 | 65.4 |
| | 30 and above | 323 | 34.6 |
| District (n = 938) | Dulkadiroğlu | 339 | 36.1 |
| | Onikişubat | 599 | 63.9 |
| Working (n = 936) | Yes | 162 | 17.3 |
| | No | 774 | 82.7 |
| Husband working (n = 935) | Yes | 885 | 94.6 |
| | No | 50 | 5.4 |
| Education level (n = 938) | Illiterate | 8 | 0.9 |
| | Literate but never attended school | 13 | 1.4 |
| | Primary school | 163 | 17.4 |
| | Secondary school | 233 | 24.8 |
| | High school | 300 | 32.0 |
| | University | 212 | 22.6 |
| | Master-doctorate | 9 | 1.0 |
| Husband's education level (n = 938) | Illiterate | 2 | 0.2 |
| | Literate but never attended school | 2 | 0.2 |
| | Primary school | 136 | 14.5 |
| | Secondary school | 207 | 22.1 |
| | High school | 296 | 31.6 |
| | University | 277 | 29.5 |
| | Master-doctorate | 18 | 1.9 |
| Living child (n = 929) | Yes | 634 | 68.2 |
| | No | 295 | 31.8 |
| Monthly income (Turkish Lira) (n = 911) | 2,500 and below | 486 | 53.3 |
| | 2,501 and above | 425 | 46.7 |
| Number of pregnancies (n = 938) | 2 and below | 533 | 56.8 |
| | 3 and above | 405 | 43.2 |
| Gestational week (n = 937) | 20 weeks and below | 377 | 40.2 |
| | 21 weeks and above | 560 | 59.8 |

**Table 2 Knowledge and attitudes of pregnant women regarding vaccination.**

| | | n | % |
|---|---|---|---|
| Vaccination of living child (n = 634) | Yes | 623 | 98.3 |
| | No | 1 | 0.1 |
| | Partial | 10 | 1.6 |
| Intention to vaccinate their future baby (n = 938) | Yes | 918 | 97.9 |
| | No | 11 | 1.1 |
| | Partial | 9 | 1.0 |

| | | n | % |
|---|---|---|---|
| Reason for not being fully vaccinated (n = 20)* | Harmful | 14 | 77.8 |
| | Not beneficial | 7 | 38.9 |
| | It causes infertility | 1 | 5.5 |
| | Lack of trust in vaccination | 11 | 6.1 |
| | Religious reasons | 4 | 22.2 |
| | Relative recommendation | 4 | 22.2 |
| | Health professionals recommendation | 1 | 5.5 |
| | Meeting someone with a side effect | 4 | 22.2 |
| | Other | 3 | 16.7 |
| Source of information on vaccination (n = 929) | Healthcare personnel | 694 | 74.7 |
| | Internet | 113 | 12.2 |
| | Friends-relatives | 69 | 7.4 |
| | Television | 43 | 4.6 |
| | Newspaper-book-magazine | 8 | 0.9 |
| | Other | 2 | 0.2 |
| Trusting the physician about vaccination (n = 931) | I don't trust | 26 | 2.8 |
| | I trust | 721 | 77.4 |
| | I am very confident | 184 | 19.8 |
| Is the vaccine free? (n = 923) | No | 13 | 1.4 |
| | Yes | 910 | 98.6 |
| Do you know where to vaccinate? (n = 931) | Yes | 915 | 98.3 |
| | No | 16 | 1.7 |
| Have you heard of herd immunity? (n = 933) | Yes | 201 | 21.5 |
| | No | 732 | 78.5 |

**Note:**
* More than one answer was provided.

**Table 3 Relationship between sociodemographic features and vaccine refusal.**

| | Total | | Pro vaccine | | Vaccine refusal | | |
|---|---|---|---|---|---|---|---|
| Parameters | n | %a | n | %b | n | %b | p |
| **Age groups (years)** | | | | | | | |
| 29 and below | 610 | 65.4 | 594 | 97.4 | 16 | 2.6 | 0.165 |
| 30 and above | 323 | 34.6 | 319 | 98.8 | 4 | 1.2 | |
| **District** | | | | | | | |
| Dulkadiroğlu | 339 | 36.1 | 333 | 98.2 | 6 | 1.8 | 0.563 |
| Onikişubat | 599 | 63.9 | 585 | 97.7 | 14 | 2.3 | |
| **Montly income (Turkish Lira)** | | | | | | | |
| 2,500 and below | 486 | 53.3 | 480 | 98.8 | 6 | 1.2 | **0.034** |
| 2,501 and above | 425 | 46.7 | 411 | 96.7 | 14 | 3.3 | |
| **Education level** | | | | | | | |
| High school and below | 717 | 76.7 | 707 | 98.6 | 10 | 1.4 | **0.012*** |
| University and above | 218 | 23.3 | 208 | 95.4 | 10 | 4.6 | |

| | Total | | Pro vaccine | | Vaccine refusal | | |
|---|---|---|---|---|---|---|---|
| Parameters | n | %a | n | %b | n | %b | p |
| **Husband education level** | | | | | | | |
| High school and below | 643 | 68.6 | 636 | 98.9 | 7 | 1.1 | **0.001** |
| University and above | 294 | 31.4 | 282 | 95.6 | 13 | 4.4 | |
| **Living child** | | | | | | | |
| Have | 634 | 68.2 | 628 | 99.1 | 6 | 0.9 | **<0.001** |
| Don't have | 295 | 31.8 | 281 | 95.3 | 14 | 4.7 | |
| **Number of children** | | | | | | | |
| 1 and below | 601 | 64.1 | 584 | 97.2 | 17 | 2.8 | **0.049** |
| 2 and above | 337 | 35.9 | 334 | 99.1 | 3 | 0.9 | |
| **Information source** | | | | | | | |
| Internet | 113 | 12.2 | 103 | 91.2 | 10 | 8.8 | **<0.001**[*] |
| Other | 816 | 87.8 | 806 | 98.8 | 10 | 1.2 | |
| **Trusting the physician about vaccination** | | | | | | | |
| No | 26 | 2.8 | 17 | 65.4 | 9 | 34.6 | **<0.001** |
| Yes | 905 | 97.2 | 894 | 98.8 | 11 | 1.2 | |

Notes:
a, Column percentage; b, Row percentage.
[*] Fisher's exact test.

**Table 4 Relationship between vaccine knowledge and vaccine refusal.**

| | Total | | Pro vaccine | | Vaccine refusal | | |
|---|---|---|---|---|---|---|---|
| Questions | n | %a | n | %b | n | %b | p |
| **If we don't vaccinate our children, rare diseases will reappear** | | | | | | | |
| True | 885 | 95.0 | 878 | 99.2 | 7 | 0.8 | **<0.001**[*] |
| False | 47 | 5.0 | 35 | 74.5 | 12 | 25.5 | |
| **The whole community benefits from vaccination** | | | | | | | |
| True | 860 | 92.1 | 852 | 99.1 | 8 | 0.9 | **<0.001**[*] |
| False | 74 | 7.9 | 62 | 83.8 | 12 | 16.2 | |
| **No need for vaccination if the baby is raised naturally and healthy** | | | | | | | |
| True | 40 | 4.3 | 26 | 65.0 | 14 | 35.0 | **<0.001**[*] |
| False | 893 | 95.7 | 887 | 99.3 | 6 | 0.7 | |
| **The diseases that the vaccine protects against are not severe. Vaccination can be avoided** | | | | | | | |
| True | 27 | 2.9 | 14 | 51.9 | 13 | 48.1 | **<0.001**[*] |
| False | 907 | 97.1 | 901 | 99.3 | 6 | 0.7 | |
| **Immediately after vaccination, a number of side effects may occur** | | | | | | | |
| True | 683 | 73.7 | 663 | 97.1 | 20 | 2.9 | **0.007** |
| False | 244 | 26.3 | 244 | 100 | 0 | 0.0 | |
| **Side effects of the vaccine appear years later** | | | | | | | |
| True | 117 | 12.8 | 102 | 87.2 | 15 | 12.8 | **<0.001**[*] |
| False | 798 | 87.2 | 793 | 99.4 | 5 | 0.6 | |

(Continued)

| Questions | Total | | Pro vaccine | | Vaccine refusal | | |
|---|---|---|---|---|---|---|---|
| | n | %a | n | %b | n | %b | p |
| **Vaccines cause autism** | | | | | | | |
| True | 56 | 6.1 | 50 | 89.3 | 6 | 10.7 | **<0.001**\* |
| False | 861 | 93.9 | 850 | 98.7 | 11 | 1.3 | |
| **Vaccines weaken the immune system** | | | | | | | |
| True | 67 | 7.3 | 56 | 83.6 | 11 | 16.4 | **<0.001**\* |
| False | 857 | 92.7 | 850 | 99.2 | 7 | 0.8 | |
| **Since the baby is small in the first 6 months, it is better to vaccinate later** | | | | | | | |
| True | 66 | 7.2 | 58 | 87.9 | 8 | 12.1 | **<0.001**\* |
| False | 847 | 92.8 | 840 | 99.2 | 7 | 0.8 | |
| **Some vaccines are more dangerous than the diseases they protect against** | | | | | | | |
| True | 65 | 7.2 | 50 | 76.9 | 15 | 23.1 | **<0.001**\* |
| False | 841 | 92.8 | 838 | 99.6 | 3 | 0.4 | |
| **Some vaccines contain heavy metals such as mercury** | | | | | | | |
| True | 189 | 21.2 | 170 | 89.9 | 19 | 10.1 | **<0.001**\* |
| False | 703 | 78.8 | 702 | 99.9 | 1 | 0.1 | |
| **Healthcare workers only talk about the benefits of vaccination** | | | | | | | |
| True | 387 | 41.6 | 369 | 95.3 | 18 | 4.7 | **<0.001** |
| False | 543 | 58.4 | 541 | 99.6 | 2 | 0.4 | |
| **I was not sufficiently informed about vaccines by health workers** | | | | | | | |
| Yes | 297 | 32.7 | 284 | 95.6 | 13 | 4.4 | **<0.001** |
| No | 612 | 67.3 | 607 | 99.2 | 5 | 0.8 | |
| **I have met someone, who have been exposed to vaccine side effects** | | | | | | | |
| Yes | 99 | 10.7 | 89 | 89.9 | 10 | 10.1 | **<0.001**\* |
| No | 822 | 89.3 | 812 | 98.8 | 10 | 1.2 | |

**Notes:**
a, Column percentage; b, Row percentage.
\* Fisher's exact test.

**Table 5 Factors affecting vaccine refusal.**

| Parameters | p | OR | 95% CI |
|---|---|---|---|
| **Age** | 0.252 | 0.90 | [0.76–1.07] |
| **Education level** | **0.033** | 0.14 | [0.02–0.86] |
| High school and below | | | |
| University and above-Ref | | | |
| **Rare diseases will come back if we don't vaccinate our children** | 0.217 | 3.22 | [0.50–20.57] |
| True-Ref | | | |
| False | | | |
| **The whole community benefits from vaccination** | **0.002** | 25.51 | [3.41–191.06] |
| True-Ref | | | |
| False | | | |

| Parameters | *p* | OR | 95% CI |
|---|---|---|---|
| **No need for vaccination if the baby is raised naturally and healthy** | **0.003** | 14.37 | [2.40–85.84] |
| True | | | |
| False-Ref | | | |
| **Vaccines cause autism** | 0.940 | 1.07 | [0.17–6.96] |
| True | | | |
| False-Ref | | | |
| **Since the baby is small in the first 6 months, it is better to vaccinate later** | **0.030** | 6.30 | [1.20–33.17] |
| True | | | |
| False-Ref | | | |
| **Some vaccines are more dangerous than the diseases they protect against** | **0.001** | 30.70 | [3.99–236.11] |
| True | | | |
| False-Ref | | | |

**Note:**
CI, Convidence interval; OR, Odds ratio.

Logistic regression analysis examined the effect of age, education level and participants' thoughts about vaccination in terms of vaccine refusal. The belief that there is no need for vaccination if the baby is raised naturally and healthily increases the risk of vaccine refusal 14.37-fold (OR 14.37 95% CI [2.40–85.84]), and participants who believe that some vaccines are more dangerous than the disease they are intended to protect, were 30.70-fold (OR 30.70 95% CI [3.99–236.11]) more likely to refuse vaccination (Table 5).

# DISCUSSION

In our study, 98.3% of pregnant mothers with a living child stated that they fully vaccinated their child, 97.9% of pregnant mothers stated that they would have all vaccinations for their child to be born, and 2.1% stated that they would have vaccinations partially or would not vaccinate their newborns at all. It is known that vaccination behaviors are shaped in pregnancy (*Glanz et al., 2013*). Therefore, it is crucial to define attitudes towards vaccine hesitancy or refusal in pregnant women and reveal the underlying reasons.

When the studies with parents were reviewed, in a study conducted in Türkiye, 1.4% of the parents refused to vaccinate their children on the grounds that vaccination was harmful or not beneficial (*Üzüm et al., 2019*). Similar to our results, 2.29% of the parents refused to vaccinate their children for various reasons in a study conducted in the Czech Republic (*Dáňová et al., 2015*). Vaccine hesitation rates were 16.3%, 24.6%, and 34.7% in three different studies conducted in Italy, respectively, while this rate was 40% in a study conducted in Canada (*Giambi et al., 2018*; *Bianco et al., 2019*; *Napolitano, D'Alessandro & Angelillo, 2018*; *Dubé et al., 2016a*). Vaccine hesitation and refusal behaviours vary by region. In a study on pregnant women, such as ours, the rate of pregnant women who said they would have the Hexavalent (HBV, Polio, Haemophilus influenza B, Tetanus, Diphtheria, Pertussis) vaccine without hesitation was 80.4%. In contrast, the rate of pregnant women who said they would have the Measles, Mumps, Rubella vaccine without hesitation was 71.1% (*Massimi et al., 2017*). Following the initial interview, approximately

half of the pregnant women in a Canadian study of 56 pregnant women were classified as hesitant about vaccination. Of these participants, 48% were experiencing their first pregnancy (*Dubé et al., 2016b*). In our study, those who had their first pregnancy made up a sizable proportion of the group who refused vaccines. This demonstrated that women experiencing their first pregnancy are inexperienced, indecisive, and may engage in risky behaviors, thus requiring special attention.

Our study investigated the relationship between vaccine refusal and sociodemographic characteristics. While there was no statistically significant relationship between maternal age and vaccine refusal, it was observed that vaccine refusal was significantly higher among pregnant women with a high income, advanced parental (maternal and paternal) educational levels, and fewer than two children. The literature has frequently compared sociodemographic characteristics and vaccine hesitancy or refusal. According to some studies, having with three or more children may either increase or decrease the rate of regular vaccination (*Üzüm et al., 2019*; *Gülgün et al., 2014*). In a study by *Topçu et al. (2019)* vaccine hesitation and refusal rates were lower in those with a high-income level and a high educational level of parents (mother and father). No statistically significant difference was observed between maternal age, number of children, and vaccine hesitation (*Topçu et al., 2019*). In a Malaysian study that compared groups with and without vaccine hesitancy, vaccine hesitancy was lower if the maternal age was more than 40 years and there were more than two children. However, no statistically significant difference was found between maternal education level, income, and vaccine hesitancy (*Azizi, Kew & Moy, 2017*). In an Italian study, vaccine hesitancy was found to be high in those with a university or higher education level and in parents with two or more children. In contrast, no significant relationship was found between maternal age and vaccine hesitancy (*Giambi et al., 2018*). A study conducted in Greece discovered that a higher level of education of the mother and father, and maternal and paternal age of more than 30 years, increased the rate of being fully vaccinated. In contrast, an increase in the number of siblings decreased the rate of being fully vaccinated (*Danis et al., 2010*). A study in the United Kingdom reported no significant relationship between vaccine hesitancy and maternal age or socioeconomic status (*Luyten, Bruyneel & van Hoek, 2019*). The relationship between sociodemographic characteristics and vaccine hesitancy and refusal differ between our study and remains inconclusive in the literature. This was thought to be due to regional and cultural differences. Nevertheless, it is concerning that vaccine hesitation or refusal is prevalent among parents with a higher level of education-at least for our region-considering the fact that, conspiracy theorists, some of whom are highly educated, are among the most vocal vaccine deniers (*World Health Organization (WHO), 2017*).

In our study, pregnant women were asked about their thoughts about vaccination to assess the reasons for vaccine hesitation or refusal behaviors. More than one-fourth of participants stated that there may be some side effects after vaccination, 6.1% stated that vaccines cause autism, and 7.2% stated that some vaccines are more dangerous than the diseases they protect against. We observed that vaccine knowledge and attitudes influenced vaccine refusal. According to the literature, the most common reasons for vaccine hesitation and refusal behaviors were a lack of trust in vaccine content and vaccine side

effects (*Üzüm et al., 2019*; *Dáňová et al., 2015*; *Giambi et al., 2018*; *Dubé et al., 2016a*; *Massimi et al., 2017*). Also, religious beliefs were reported to affect vaccination behaviors (*Wombwell et al., 2015*). The fact that parents are fearful of vaccine side effects and that common misconceptions influence vaccination behaviors revealed the importance of the information they received from health professionals in the aforementioned studies. In our study, 41.6% of pregnant women said health professionals only mentioned the benefits of vaccination, and 32.7% said they did not get enough information. In 74.7% of pregnant women, healthcare professionals were the source of vaccination information, while the Internet was used by 12.2% of pregnant women. In our study, those who refused vaccines were significantly more likely to use the Internet as a source of information. It is known that parents mostly receive information regarding vaccination from healthcare professionals (*Gellin, Maibach & Marcuse, 2000*). In a study evaluating the knowledge, attitudes, and practices of midwives regarding COVID-19 vaccination, it was reported that only 51.9% of midwives routinely provided information about COVID-19 vaccination to pregnant women, and obtaining information from official government organizations or scientific journals had a positive impact on midwives' practices (*Miraglia del Giudice et al., 2023*). In this regard, healthcare professionals, who serve as a source of information, should also be knowledgeable about vaccines. However concerns arise when the Internet is used as the primary source of knowledge. Similar studies reported that a high rate of internet use as a source of information is associated with vaccine hesitancy or refusal. Furthermore, in one of these studies, more than half of the parents wanted to learn more about childhood vaccines (*Giambi et al., 2018*; *Napolitano, D'Alessandro & Angelillo, 2018*). According to one study, when the words vaccination and immunization were entered into the Google search engine, 43% of the websites returned anti-vaccination content (*Davies, 2002*). Another study found that an internet-based intervention improved undecided parents' attitudes toward vaccines (*Daley et al., 2018*). Our study and other studies discovered that parents' trust in their physicians and adherence to their recommendations positively affected vaccination behaviors (*Giambi et al., 2018*; *Napolitano, D'Alessandro & Angelillo, 2018*; *Wang et al., 2021*).

Our study has some limitations. One of the limitations of the present study is that, due to time and resource limitations, the study was carried out only on the selected sample, which can lead to sampling and self-selection biases. Secondly, as it is a survey study, recall bias may affect the questionnaire responses. Another significant limitation is the susceptibility to social desirability bias. Participants may feel inclined to respond in a socially acceptable or desirable manner rather than providing genuine or truthful answers. Some of the aforementioned biases were mitigated by utilizing a large sample size and weighting this sample based on the population distribution in the research area.

On the other hand, we believe our article will provide various epidemiological data on vaccine refusal and shed light on future studies. As far as we know, such studies like ours are scarce in the literature. Most studies are conducted on those who readily refused to vaccinate their children, examining the rationale behind the refusal, which is far from representing the picture of general population. Our study has a large sample size and is population-based by design. Thus we are confident that our sample is representative of

pregnant women in Kahramanmaraş, and, to some extent, of those in Türkiye. Another strength of our study is that it was conducted among pregnant women who will make the most critical decisions regarding vaccinating their babies.

As a result, the study found that 2.1% of the pregnant participants in the study group opted to refuse vaccination either fully or partially to their unborn babies. While this rate may seem low, considering the increasing prevalence of anti-vaccine campaigns and misinformation in the digital sphere, it is deemed essential to monitor the issue carefully. In conclusion, the most common reasons for vaccine refusal are false beliefs held by parents, extensive use of the Internet as a source of information, and a lack of adequate information from health professionals. Additionally, higher-income families with fewer children, who lack trust in healthcare professionals, exhibited a more pronounced vaccine-resistant attitude. It is concluded that providing information to parents about vaccination by health professionals and using pro-vaccine internet policies as a mass communication tool could effectively counter vaccine refusal.

### Funding
The authors received no funding for this work.

### Competing Interests
The authors declare that they have no competing interests.

### Author Contributions
- Ramazan Azim Okyay conceived and designed the experiments, performed the experiments, analyzed the data, prepared figures and/or tables, authored or reviewed drafts of the article, and approved the final draft.
- Erhan Kaya performed the experiments, analyzed the data, prepared figures and/or tables, authored or reviewed drafts of the article, and approved the final draft.
- Ayşegül Erdoğan performed the experiments, analyzed the data, prepared figures and/or tables, authored or reviewed drafts of the article, and approved the final draft.

### Human Ethics
The following information was supplied relating to ethical approvals (*i.e.*, approving body and any reference numbers):

The study was approved by the Ethics Committee of Kahramanmaraş Sütçü İmam University with the date of October 24, 2018 and number 21.

### Data Availability
The raw data are available in the Supplemental File.

### Supplemental Information
Supplemental information for this article can be found online at http://dx.doi.org/10.7717/peerj.17409#supplemental-information.

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
