# Peer review of "Vaccine refusal in pregnant women in Kahramanmaraş: a community-based study from Türkiye"

_PeerJ, doi:10.7717/peerj.17409_

## Round 0.1 · original submission · Major Revisions

Reviewers raised several concerns about the design, methods, and results. These concerns need to be thoroughly addressed by the authors before the manuscript can be further evaluated for publication.

Reviewer 1 ·

Basic reporting

Materials and Methods
a) It is indicated that the study was conducted in family health centers. It is not described how many family health centers are in geographic area.
b) It is necessary to indicate how many individuals have been selected.
c) It is necessary to specify who has conducted the face-to-face interview.
d) It is not stated whether the participant was informed about the use and anonymization of the data and that survey responses guarantee the anonymity of each participant.
e) It is not given any information whether participants were not able to continue to the next question of the questionnaire if they failed to provide a response to an item.
f) It should be clarified whether the participants have received any gift or monetarily compensated.
g) It should be clarified whether a pilot study has been conducted.
h) The Authors should describe the survey questionnaire items.
i) The Authors should clarify about the face-validity testing of the questions with an explanation of the validity of the content of the questions with regard to the research aims. The Authors should clarify how they had estimated the reliability, or internal consistency, of the questions by using, for example the Cronbach’s alpha in order to measure whether or not a score is reliable.
j) One of major weakness is that the statistical analysis is not, strictly speaking, adequate, because it would be particularly relevant to describe the model(s) developed and the outcome(s), the variables included and the rationale why they are included to measure the associations between several characteristics and the outcomes of interest. Moreover, it should be indicated if the tests were one-side or two-sided.
Results
a) The response rate should be reported.
b) No information is given about those patients who refused to participate. Was there any attempt to quantify the response bias: information about non-responders. It would be useful to have some kind of indication of comparability with non-respondents. Is there any population-based data available? How did they differ from those in the sample, how representative is the sample and were the findings representative of Turkey?
Discussion
a) The pivotal role of midwives and of healthcare providers as sources of information with a positive impact towards vaccination attitudes and uptake should be stressed and studies supporting this statement should be added. For example, the following articles should be cited Miraglia del Giudice et al. Vaccines 2023;11(2): 222; Wang et al. Vaccines (Basel) 2021;9(3):29.
b) There is a lack of comparison with the results of recent studies conducted among pregnant women regarding the COVID-19 vaccination in other geographic areas. The work should therefore be enriched in such a way as to become self-supporting by photographing the context and what is around it in order to make comparisons. For example, the following article should be cited Saitoh et al. Hum Vaccin Immunother 2022 Nov 30;18(5):2064686; Germann et al. BJOG 2022 Apr 16:10.1111/1471-0528.17189; Miraglia del Giudice et al. Front Public Health 2022;10:995382.

c) The paragraph regarding the limitations of the study should discuss all limits such as, for example, the study design, the recall bias, and the social desirability bias.
Tables and Figures
a) In Table 5 report only two decimals.
References
a) The manuscript is not well referenced. The References list is not updated, since several articles conducted in different countries and published on peer-reviewed journals have been not included.

Experimental design

See comments in Basic design

Validity of the findings

See comments in Basic design

Reviewer 2 ·

Basic reporting

The article should be revised by a fluent English speaker.
The article should be written in the same design (passive or active)For ex: We. did........ or It was done........

Experimental design

It was stated that the number of participants in the study was 938. But in tables total number is different in each questions. So can you explain why the total numbers were changed. If there were unanswered questions in the questionnaires it would be better to extract those questionnaires from the study.
Do you have any permission to do the study in FHCs.? If there is please state it in the manuscript.
How many FHCs are in Kahramanmaras and in which of them did you conduct the study?

Validity of the findings

In the tables p values should be written with using dot, not comma.
Please state the strengths of your study.

Additional comments

Please give the most important findings in the abstract.
"In addition, 2.8% of pregnant women lacked trust in their physicians." This sentence should be removed.

Reviewer 3 ·

Basic reporting

Good clear straightforward reporting.
The literature review is enough and up to date
Some minor typos (attached)

Experimental design

Clear setting, enough sample size
Clear recruitment plan.
Clear operational definition of study variables.

Validity of the findings

Valid findings and appropriate statical tests.
However, conclusion does not match with results.

Hesitancy rate is very low <3% , so this showed good attitude and knowledge.

Annotated reviews are not available for download in order to protect the identity of reviewers who chose to remain anonymous.

---

## Round 0.2 · Minor Revisions

Please, address reviewer #2 comments.

Reviewer 1 ·

Basic reporting

No comment

Experimental design

No comment

Validity of the findings

No comment

Additional comments

The Authors have satistactorily addressed all concerns that have been raised.

Reviewer 2 ·

Basic reporting

Thank you for your illuminating responses to the reviewers' comments. But it is still needed minor revisions.

In the "Abstract" section the information about where the study was conducted should be added.
The article should be revised by a fluent English speaker and the grammatical errors should be revised.
"Turkey" should be changed as "Türkiye".

Experimental design

No comment.

Validity of the findings

No comment

Additional comments

The authors have revised the manuscript due to the comments. But needs some revisions for grammatical errors.

---

## Round 0.3 · accepted · Accept

Thank you for addressing the reviewer's comments. Your manuscript is now deemed suitable for publication